# Evaluation of Regulating and Provisioning Services Provided by a Park Designed to Be Resilient to Climate Change in Bangkok, Thailand

Arerut Yarnvudhi [1,2], Nisa Leksungnoen [1,2,*], Pantana Tor-Ngern [3], Aerwadee Premashthira [4], Sathid Thinkampheang [1,5] and Sutheera Hermhuk [6]

1 Department of Forest Biology, Faculty of Forestry, Kasetsart University, Bangkok 10900, Thailand; psdaryv@ku.ac.th (A.Y.); kawlica_70@hotmail.com (S.T.)
2 Center for Advance Studies in Tropical Natural Resources, Kasetsart University, Bangkok 10900, Thailand
3 Department of Environmental Science, Faculty of Science, Chulalongkorn University, Bangkok 10330, Thailand; pantana.t@chula.ac.th
4 Department of Agricultural and Resource Economics, Faculty of Economics, Kasetsart University, Bangkok 10900, Thailand; fecoadu@ku.ac.th
5 Cooperation Centre of Thai Forest Ecological Research Network, Kasetsart University, Bangkok 10900, Thailand
6 Faculty of Agricultural Production, Maejo University, Chiang Mai 50290, Thailand; h.sutheera@gmail.com
* Correspondence: ffornsl@ku.ac.th; Tel.: +66-2-5790176

**Abstract:** Understanding the ecosystem services provided by urban green spaces, in terms of their environmental, economic, and social benefits, is essential for a better management of area. Chulalongkorn University Centenary Park (CU 100) was established to mitigate the effects of climate change, especially flood prevention. This study focused on quantifying the ecosystem services provided by the trees in the park in terms of regulating and provisioning services. A publicly available tool, the i-Tree Eco international software, was used with data obtained from a local weather station as proxies to determine the accuracy of the analysis. Services, quantified in terms of monetary value, included avoided runoff, carbon storage, carbon sequestration, pollution removal, and timber price. The total monetary benefits, obtained from 697 trees (56 species, 49 genera, and 22 families), were estimated at USD 101,400. Of the total services, provisioning services contributed 75% to the total monetary value. Among all regulating services, the avoided runoff contributed about 60%, which was considered as the goal achieved by the park design. *Azadirachta indica* A. Juss (USD 518.75/tree$^{-1}$/year$^{-1}$), *Shorea roxburghii* G. Don (USD 417.17/tree$^{-1}$/year$^{-1}$) and *Millettia leucantha* Kurz (USD 414.87/tree$^{-1}$/year$^{-1}$) provided the greatest benefit, as indicated by a high value of provisioning services in terms of a high timber quality. These results can be used when planning the composition of trees to be planted in urban areas to increase both green spaces and maximize ecosystem services to improve the vitality of human well-being.

**Keywords:** i-Tree Eco international; monetary value; carbon storage and sequestration; pollution removal; avoided runoff

## 1. Introduction

Urbanization has been rapidly increasing in every continent of the world, especially Asia and Africa [1]. Over half of the world's population resides in cities, and this proportion is expected to reach 66% by 2050 [2]. With this continuous increase of urbanization [3], ecosystem services (ES) provided by urban green spaces are essential for overall human development [4]. According to the Millennium Ecosystem Assessment (MA) (2005) [5] and The Economics of Ecosystems and Biodiversity (TEEB) (2010) [6], ES have been divided into four categories: provisioning services, regulating services, cultural services, and supporting services. Provisioning services, such as food, materials, and energy, are directly used by

people. Regulating services are the processes through which the ecosystem regulates the environment, such as the microclimate, water purification, erosion and flood control, and carbon storage. Cultural services are the non-material benefits obtained by people through spiritual enrichment, recreation, and cognitive development. Supporting services are necessary for the other three services and for the production of all other ecosystem services, for example, soil formation, nutrient cycling, and the habitats of organisms [7].

In general, urban residents appreciate the cultural services provided by green areas, especially recreational activities. Most people are familiar with provisioning services, given the direct use of resources such as fishing for consumption and wood for building houses. However, awareness about the regulating and supporting services, such as nutrient cycling, is much less widespread, and therefore, these services are mostly taken for granted [8]. Moreover, an evaluation of ES is quite difficult because most of the services are intangible assets. However, it is essential to quantify the value of ES in monetary terms, as this is the most comprehensible way for people to understand and appreciate the ES [9].

Although, in an urban park ecosystem, the calculation of provisioning services related to timber value is relatively straightforward [10], other services are mostly intangible [11]. Several methods have been developed to evaluate the monetary value of natural resources. For example, carbon concentration was estimated through carbon storage, carbon flux, or eddy covariance [12,13], with the monetary value calculated using the carbon market price [14]. However, some ES are still challenging to evaluate due to the complexity of techniques such as surface runoff prevention, pollution removal, and oxygen production. Recent developments of the models have helped with the calculation and estimation of ES. A few examples are ARIES, Co$ting Nature, LUCI, InVEST, Water Word, and i-Tree [15]. However, some models have focused on mapping and land use outside urban areas, such as ARIES, Co$ting Nature, and LUCI. While some models are available online and are free, such as InVEST and i-Tree Eco, the level of complexity varies for each model. For example, InVEST provides a suite of 18 models that can be used as tools to estimate the value of different services [15].

The i-Tree Eco model has been developed over the past 20 years for use in urban forests [16]. The model estimates the carbon storage, carbon sequestration, air pollution removal, and avoided runoff, based on the composition and structure of local vegetation, and then the estimations are converted to a monetary value [17]. Originally, the model was used for plant species growing in the temperate zone of the U.S.A. [18]. The model has subsequently been extended to include Canada [19] and Mexico [20], and to estimate the ES in these neighboring countries using climate data from the U.S.A. Other countries have also used this software with their local weather data, such as Australia [21], Brazil [22], and some European countries [23], and with similar plant species. As the model is parameterized using local vegetation composition and relies on local climate data, which can highly influence the estimated ES and their monetary value [17], the i-Tree Eco has some uncertainties due to the non-availability of local weather data. As a result, the estimates are reliable in areas with climates similar to the U.S.A. [24]. For example, in Thailand, where the climate is classified as tropical, the weather data used by Choothong et al. [25] was from Hawaii. Lin et al. [26] suggested that it is best to use the available local environmental data to minimize the uncertainty resulting from the input data. Later, an international database was developed in the i-Tree Eco model to include other countries with different climates and plant species from the U.S.A. [27]. Some studies in Asia have reported the use of the i-Tree International [28–32]. Our investigation is the first of its kind undertaken in Thailand that quantifies the monetary value of regulating services using the i-Tree Eco International model of each tree species. The accuracy of estimation would increase due to the use of specific allometric equations and weather data obtained from local stations.

Given the need for expanding green spaces in cities for urban residents, many projects involving the establishment of new parks [31] have been undertaken around the world [33]. In Thailand, the green spaces in most big cities are still less than the recommended minimum area per capita, based on the UN standard (30 m$^2$/capita$^{-1}$). For instance, the green

space in Bangkok covers 6.70 m$^2$/capita$^{-1}$, 3.96 m$^2$/capita$^{-1}$ in Nakhon Si Thammarat, and only 3.36 m$^2$/capita in Chiang Mai [34]. This has led to prioritized planning to increase the area of urban green spaces through programs such as the Green Bangkok 2030 project, which aims to increase the area of green spaces to 10 m$^2$/capita$^{-1}$ by the year 2030 [34]. In accordance with this plan, a newly designed park, named "Chulalongkorn University Centenary Park" or "CU 100 Park", was established in the center of Bangkok in 2016 to promote green spaces over an area of 4.48 ha, and is surrounded by business buildings and Chulalongkorn University. The CU 100 Park was intentionally designed to reduce the impact of climate change, especially to prevent floods similar to the Megaflood of 2011, which caused tree mortality and property damage [35].

However, questions about the breakeven costs or the worthiness of creating such green areas, compared to building business centers, are still being debated. Most people would only evaluate tangible assets, when intangible value, hidden in the ES, is equally important. Therefore, this study aimed to assess the regulating and provisioning services of the CU 100 Park and to estimate their monetary value to indicate the worthiness of having green areas for urban inhabitants. We hypothesized that the regulating services might provide more monetary value, relative to the provisioning services, due to the complexity of the services provided. In addition, we would like to validate the flood prevention design outlined in the regulating services of the park via its monetary value for its avoided runoff service. The outcome of this study can benefit urban planners and select suitable tree species that are of high monetary value, in terms of regulating and provisioning services, for the expansion of green spaces in Bangkok.

## 2. Materials and Methods

### 2.1. Study Area

The study was conducted in the Chulalongkorn University Centenary Park, located in the heart of Bangkok, Thailand (long 13°44′22″ N, 100°31′25″ E) (Figure 1). The park is located within the property of Chulalongkorn University, Pathum Wan District, Bangkok, Thailand, near Chao Phraya River, at an elevation of 2 m above mean sea level and with a total area of 4.48 ha [36]. Bangkok has a seasonal monsoonal climate, where the high average daily temperature remains relatively constant over the year, largely fluctuating within the range of 28–30 °C. The climate is tropical, with the wet season receiving a mean annual rainfall between 1400–1600 mm between May and October, while the dry season spans from November to April [36].

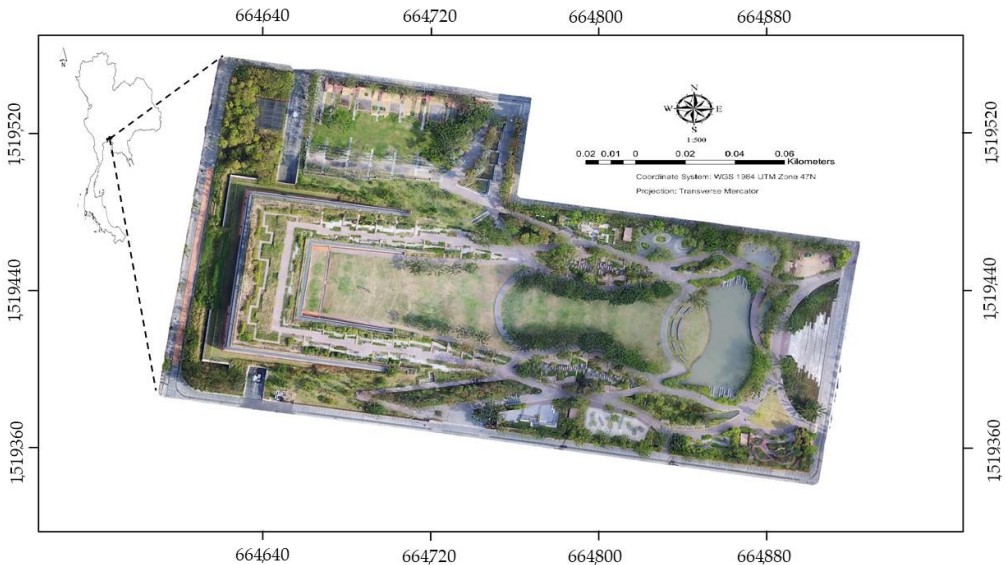

**Figure 1.** Location of the Chulalongkorn University Centenary Park (CU 100) in the Bangkok metropolitan area with an area of 4.48 ha.

The park was designed to cope with the future uncertainties of climate change, especially floods, which are mitigated through a declining slope (west–east) to allow the excess rain water to flow into a reservoir (located on the east side of the park) and through underground water drainage [37]. With this unique design, the CU 100 Park won the World Landscape Award (WLA) in 2019, organized by the World Landscape Architectural Design, China [38].

The park was established in 2016, on a previous junk yard with shops that mostly sold car parts in the center of a business area and in accordance with the Green Bangkok Project [39], to increase the area of green spaces in the Bangkok metropolitan area. The green roof was intentionally designed to be tilted at a slope of 3° in order to drain the water into the retention pond below, with the addition of underground drainpipes that remove excess water during flooding.

*2.2. Plant Inventory and Measurements*

All trees in the park, with a diameter at breast height (DBH) greater than or equal to 4.5 cm, and a height greater than 1.30 m, were measured from September to October 2019 with a diameter tape (Model 283D, Forestry Suppliers, Inc., Jackson, MS, USA) and a rangefinder (Nikon forestry Pro, Nikon Inc., Tokyo, Japan), respectively. The species were identified by experts at the Department of Forest Biology, Faculty of Forestry and Department of Botany, Faculty of Science, and the herbarium specimens from the Forest Herbarium, Department of National Parks Wildlife and Plant Conservation, Bangkok, Thailand [40].

2.2.1. Evaluating the Monetary Value of Regulating Services Provided by Trees Using the i-Tree Eco International Model Version 6 (V6)

The i-Tree Eco model provides estimates of the monetary values of carbon, carbon sequestration, avoided runoff, and pollutant removal for each species. The model was developed by the United States Department of Agriculture (USDA) for urban trees [41]. We chose the complete inventory project type for this study, in which all the trees classified above were measured individually. Field measurements were conducted according to the i-Tree field manual [27] and included (1) plot information (% tree cover), (2) individual tree attributes (species, stem diameter, height, crown size, missing crown canopy and dieback, crown light exposure, and distance and direction from a building), (3) weather data, and (4) air pollution data. The parameters used in the model are presented in Table 1. All the measurements and estimations were completed through a visual inspection by 8 trained crews (following the i-Tree protocol).

Carbon Storage and Sequestration

The model calculates carbon storage and sequestration based on the trees measured for each species. Carbon storage is the amount of carbon bound up in the above-ground and below-ground parts of woody vegetation. Carbon storage is the carbon accumulated in the total dry biomass of an individual tree. Dry biomass was calculated using allometric equations reported in the literature, which have previously been used with the i-Tree program [42]. Open-grown, maintained trees tend to have a lower biomass than predicted by forest-derived biomass equations [18]; thus, the total dry biomass was multiplied by a factor of 0.8 for urban trees. Carbon storage was then calculated by multiplying a factor of 0.5 (average % of carbon in plant tissue) by the total dry biomass.

Carbon sequestration is the removal of carbon dioxide from the air by plants. To estimate the gross amount of carbon sequestered annually, the tree's average diameter and its overall condition (health condition and crown light exposure) were used to estimate the tree diameter and the carbon to be stored during the next year. The annual carbon sequestration was calculated as the difference between the carbon storage during the current and the next year [18,43]. The monetary estimation of carbon storage and sequestration was based on the carbon value for the U.S.A. [44] (Table 1).



**Table 1.** Model input data and parameters used in i-Tree Eco for the estimation of the monetary value of regulating services in the CU 100 Park.

| Model | Input Data/Parameter | Value/ID/Monitor * | Data Year | Reference |
|---|---|---|---|---|
| | Weather data [a] | 484500: Bangkok | 2017 | NCEI, 2017 |
| | | 484500: Bangkok | 2017 | NCEI, 2017 |
| | | 484500: Bangkok | 2017 | NCEI, 2017 |
| | | 484500: Bangkok | 2017 | NCEI, 2017 |
| Carbon Storage/ Sequestration | Social cost of carbon [a] | USD 210.63 per metric ton | 2015 | IWG, 2015 |
| Air Pollutant Removal | CO concentration [b] Hourly average = 0.60 ppm (threshold = 3.492 ppm) | USD 1628.66 per metric ton | 2017 | KU Station, 2017 |
| | $NO_2$ concentration [b] Hourly average = 11.57 ppb (24 h = 13 ppb) | USD 11,466.82 per metric ton | 2017 | KU Station, 2017 |
| | $O_3$ concentration [b] Hourly average = 12.31 ppb (8 h $\leq$ 51 ppb) | USD 11,466.82 per metric ton | 2017 | KU Station, 2017 |
| | $PM_{2.5}$ concentration [b] Hourly average = 16.51 μg/m$^3$ (24 h = 15 μg/m$^3$ | USD 7655.87 per metric ton | 2017 | KU Station, 2017 |
| | $SO_2$ concentration [b] Hourly average = 1.18 ppb (24 h = 15 ppb) | USD 2807.28 per metric ton | 2017 | KU Station, 2017 |
| Avoided runoff | Stormwater control cost [c] | USD 2.5 per m$^3$ | 2017 | Vargas and others, 2007 |

[a]: Globally applicable data/parameter. [b]: Replaceable with local data (measured 17 km away from the CU 100 Park)/parameter used from the i-Tree Database; also replaced in this study [45]. [c]: Parameter used in the U.S.A. and employed in this study. * each customized i-Tree model was run using a general value for which the local values were not measured; the estimates were based on the value of the U.S.A. metro station ID [45,46] and the WHO [47].

Air Pollution Removal

The estimates of air pollution removal were based on hourly values of tree-canopy resistance to air pollution using a hybrid model of big-leaf and multi-layer canopy deposition [48]. Input data for the model included weather data from the national climatic data center (NCEI #484540). The model was optimized to run on the local levels of measured air pollution at the Kasetsart University station in Bangkok, Thailand (17 km away from the study site) (Table 1) [45]. The tree structure, including the tree cover, the percentage of evergreen trees, and the leaf area index (LAI) (which was calculated through the i-Tree Eco using crown cover data) were the key parameters for estimating the amount of air pollution removed [49,50].

Avoided Runoff

The annual avoided runoff was calculated based on the rainfall intercepted by the vegetation, specifically, the difference between the estimated annual runoff with and without vegetation. Although the tree leaves, branches, and bark may intercept precipitation and thus mitigate surface runoff, only the precipitation intercepted by the leaves was accounted for in the present analysis. Estimates were generated based on the current conditions with and without trees in order to estimate the impact of trees on the surface runoff. Impervious cover beneath the trees was assumed to be 25.5%, based on the averaged USDA impervious cover [51]. The soil information is not available in the i-Tree Eco software; therefore, it was assumed that all of the rainwater reaching the pervious cover infiltrates into the ground. The default value of rainwater runoff from one impervious cover to the next was assumed to be 1.5 mm, as found by Wang et al. [52]. The pervious cover, including grass, was assumed to be similar to that of the study based on the U.S. Forest Service's Community Tree Guide Series [53]. The avoided runoff was quantified using the value provided in the i-Tree Eco software, which was USD 2.5/m$^{-3}$ for stormwater control facilities in the U.S.A.



2.2.2. Provisioning Services Based on Timber Market Price

The monetary benefit for each tree species was calculated using the local market value provided by the Bank for Agriculture and Agricultural Cooperative (BAAC) (2018) [54] according to the mortgage securities of the tree planning project. The calculation was based on the allometric equations determining the relationship between the biomass and DBH for each species. In addition to the calculated volume, the growth rate (fast-growing vs. slow-growing species and wood type (hardwood vs. softwood)) [55] were used for estimating the monetary value of timber in the BAAC program. Fast-growing hardwood species of similar size provided a greater monetary value than slow-growing softwood species.

## 3. Results

The plant information and characteristics of the measured trees, along with the monetary value associated with the four regulating services and one provisioning service (based on timber value), is presented in the Supplementary Table S1. The survey undertaken in this park included a total of 697 individual trees, which were classified into 56 species in 49 genera and 22 families, with 76% classified as hardwood and 24% as softwood. In terms of leaf phenology, 61% species were deciduous, whereas 39% species were evergreen. Four regions of habitat were identified for these species, including all parts of America, the Middle East, Asia, Australia, and Africa. The most frequently recorded families in the area were Fabaceae (10 species), followed by Combretaceae (5 species), Bignoniacae (4 species), Moraceae (4 species), and Malvaceae (4 species). The top five abundant species were *Dalbergia cochinchinensis* Pierre (19.7%), *Tabebuia rosea* (Bertol.) DC. (11.6%), *Albizia saman* (Jacq.) Merr. (10.6%), *Millingtonia hortensis* L.F. (10.0%), and *Dipterocarpus alatus* Roxb. Ex G. Don (6.7%).

The total monetary benefits from both the regulating and provisioning services were estimated around USD 101,400.60, with an average of USD 145.48/tree$^{-1}$/year$^{-1}$ (Table 2 and Figure 1). Based on the estimation from the model, the contribution of timber value to the provisioning service was 75%, while the remaining 25% was contributed by the regulating services. The five species with the highest value per tree$^{-1}$ per year$^{-1}$ were *Azadirachta indica* A.Juss, *Shorea roxburghii* G.Don, *Millettia leucantha* Kurz, *Xylia xylocarpa* (Roxb.) Taub. Var. *kerrii* (Craib & Hutch.) I.C.Neilsen, and *D. cochinchinensis*, with an estimated monetary value of USD 518.75, USD 417.17, USD 414.87, USD 249.20, and USD 248.82/tree$^{-1}$/year$^{-1}$ (Table 2), respectively.

The CU 100 Park was specifically designed to counter the ill effects of climate change and also regulate the microclimate around the park. The design intentionally prevented flooding and water runoff resulting from soil compaction and impervious cover, such as cement, concrete, and asphalt material. Among all the regulating services, up to 60% contributed to avoided runoff, followed by carbon storage (26%), with oxygen production, carbon sequestration, and pollution removal together contributing 14% (Figure 2). The results suggest that the design of the park—primarily to control flooding—was achieved. Trees in the park were estimated to help mitigate a total of 81.99 m$^3$ of water runoff (18.30 m$^3$/ha$^{-1}$), with an associated value of USD 211 per year$^{-1}$. The main tree traits contributing the most to avoided runoff were tree size and crown canopy, especially the latter, which was strongly correlated with runoff avoidance (correlation coefficient $^®$ = 0.851, $p < 0.001$). The tree species that contributed the most in terms of avoided runoff in this park included *M. leucantha, A. saman, Gliricidia sepium* (Jacq.) Walp., *S. roxburghii, Sterculia gilva* Miq., *Terminalia arjuna* (Roxb. Ex DC.) Wight & Arn.), and *Saraca indica* L.

**Table 2.** Combined monetary benefits of the regulating and provisioning services in the Chulalongkorn Centenary Park (CU 100 Park).

| Species Name | Forest Type | Habitat | DBH (cm) | Total Height (m) | Tree Count | Monetary Value from i-Tree Eco Calculation (USD) (1) | | | | | | | | | Monetary Value from BAAC (USD) (2) | | Total Monetary Value (USD) (1) + (2) |
| | | | | | | Carbon Storage | | Gross Carbon Sequestration | | Avoid Runoff | | Pollution Removal | | Oxygen | Timber Value | | |
| | | | | | | kg tree$^{-1}$ year$^{-1}$ | USD tree$^{-1}$ year$^{-1}$ | kg tree$^{-1}$ year$^{-1}$ | USD tree$^{-1}$ year$^{-1}$ | m$^3$ tree$^{-1}$ year$^{-1}$ | USD tree$^{-1}$ year$^{-1}$ | kg tree$^{-1}$ year$^{-1}$ | USD tree$^{-1}$ year$^{-1}$ | kg/tree | Total (USD) | USD tree$^{-1}$ year$^{-1}$ | USD tree$^{-1}$ year$^{-1}$ |
|---|---|---|---|---|---|---|---|---|---|---|---|---|---|---|---|---|---|
| *Azadirachta indica* | D | NA | 18.80 ± 0.96 | 8.30 ± 0.83 | 3 | 67.06 | 13.57 | 12.33 | 2.48 | 0.17 | 0.43 | 1.29 | 14.54 | 32.90 | 1463.18 | 487.73 | 518.75 |
| *Shorea roxburghii* | D | NA | 14.45 ± 9.55 | 6.15 ± 2.45 | 2 | 57.60 | 11.74 | 12.40 | 2.49 | 0.30 | 0.76 | 2.27 | 25.58 | 33.00 | 753.19 | 376.60 | 417.17 |
| *Millettia leucantha* | D | NA | 16.85 ± 1.15 | 8.45 ± 0.75 | 2 | 57.95 | 11.67 | 7.75 | 1.56 | 0.37 | 0.93 | 2.77 | 31.23 | 20.70 | 738.95 | 369.48 | 414.87 |
| *Xylia xylocarpa* | D | NA | 13.74 ± 2.33 | 8.47 ± 1.22 | 17 | 33.82 | 6.84 | 7.80 | 1.57 | 0.17 | 0.36 | 1.07 | 12.11 | 20.81 | 3881.34 | 228.31 | 249.20 |
| *Dalbergia cochinchinensis* | D | NA | 13.82 ± 2.11 | 8.33 ± 1.43 | 136 | 36.18 | 7.28 | 11.23 | 1.29 | 0.13 | 0.32 | 0.97 | 10.86 | 17.12 | 31,152.70 | 229.06 | 248.82 |
| *Hopea odorata* | D | A | 13.68 ± 2.14 | 8.70 ± 1.37 | 26 | 35.66 | 7.19 | 6.27 | 1.21 | 0.05 | 0.13 | 0.39 | 4.41 | 16.72 | 5986.74 | 230.26 | 243.26 |
| *Chukrasia tabularis* | E | NA | 13.49 ± 3.12 | 7.69 ± 0.90 | 9 | 35.47 | 7.15 | 7.13 | 1.43 | 0.04 | 0.14 | 0.42 | 4.72 | 19.01 | 2048.94 | 227.66 | 241.10 |
| *Terminalia arjuna* | D | NA | 12.57 ± 3.58 | 5.97 ± 2.52 | 3 | 27.20 | 5.58 | 8.37 | 1.69 | 0.27 | 0.68 | 2.05 | 23.09 | 22.30 | 608.18 | 202.73 | 233.77 |
| *Afzelia xylocarpa* | D | NA | 13.54 ± 2.03 | 7.26 ± 0.83 | 27 | 43.56 | 6.95 | 5.89 | 1.18 | 0.04 | 0.11 | 0.34 | 3.59 | 15.69 | 5938.30 | 219.94 | 231.77 |
| *Terminalia alata* | D | NA | 13.34 ± 1.42 | 7.06 ± 1.07 | 5 | 29.60 | 5.69 | 6.20 | 1.25 | 0.08 | 0.21 | 0.64 | 7.19 | 16.50 | 1033.55 | 206.71 | 221.31 |
| Other species | D, E | A, NA, ME, OC | 12 ± 4.21 | 6.43 ± 1.60 | 467 | 1628.63 | 332.48 | 284.48 | 80.65 | 4.86 | 11.79 | 48.62 | 391.25 | 775.19 | 30,926.49 | 2224.13 | 3040.30 |
| **Total** | D, E | A, NA, ME, OC | 12.63 ± 3.98 | 6.64 ± 1.58 | 697 | 2052.73 | 416.42 | 363.65 | 96.87 | 6.48 | 15.86 | 60.83 | 528.58 | 989.94 | 84,531.57 | 5002.60 | 6060.32 |

Forest type: "D" = Deciduous, "E" = Evergreen. Habitat: "A" = Central/North/South America, "NA" = native to Thailand, "ME" = Middle East, and "OC" = other continents (Australia and Africa).

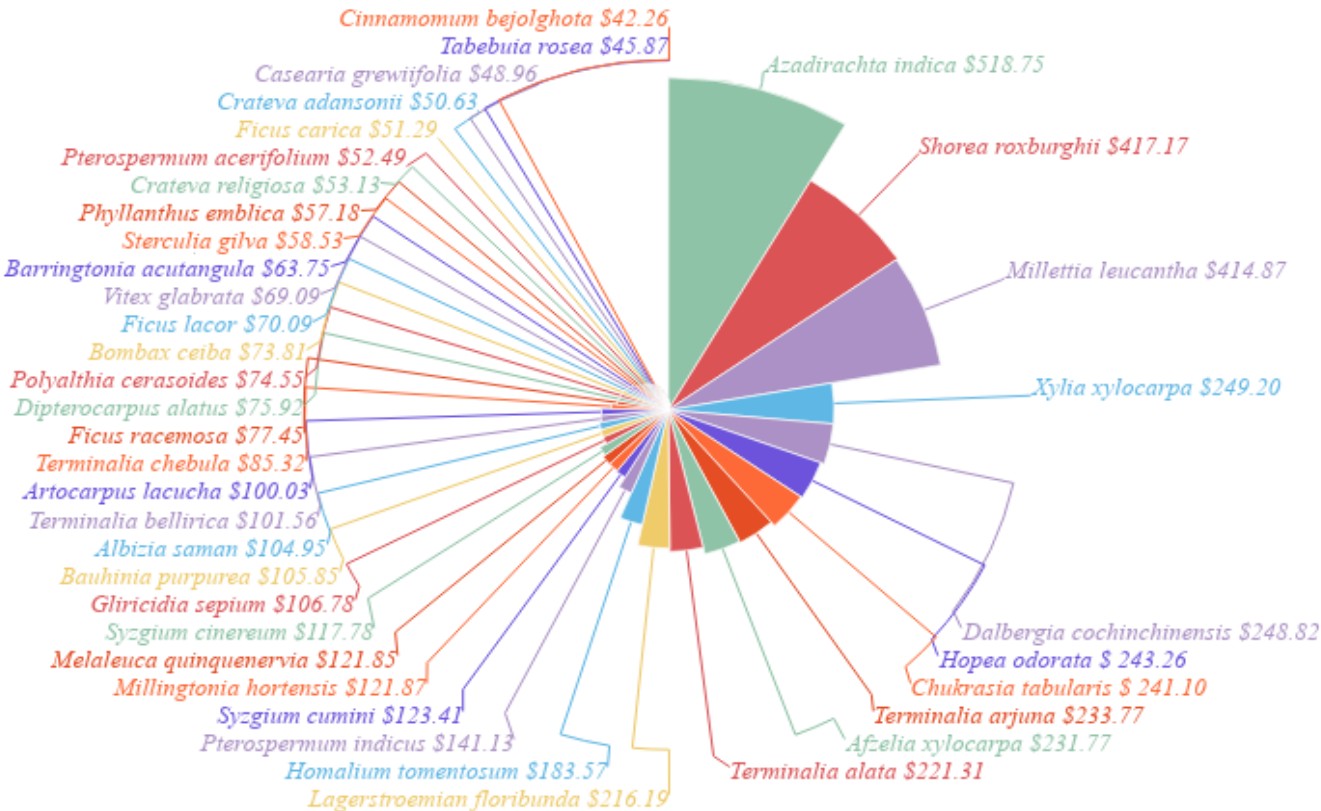

**Figure 2.** Total monetary benefit (USD/tree$^{-1}$/year$^{-1}$) of the 56 tree species identified in the park (See Supplementary Table S1 for more details on each species).

The model estimated that the trees stored 35,120 kg of carbon per year$^{-1}$ (7839.29 kg/ha$^{-1}$) and sequestered 5050 kg of carbon per year$^{-1}$ (1127.23 kg/ha$^{-1}$), amounting to USD 7200 and USD 1036 per kg per year$^{-1}$, respectively. Carbon sequestration was affected by above-ground biomass, which depended on the tree's size and health. The majority of planted trees were species native to the tropics area, including *Ficus lacor* Buch.-Ham., *S. roxburghii*, *A. indica*, and *G. sepium*, while a species native to Central and South America, *A. saman*, was also recorded.

The model shows that the trees removed approximately 619.10 kg of pollutants per year$^{-1}$ (138.20 kg/ha$^{-1}$), amounting to approximately USD 7133 per year$^{-1}$. A positive correlation was found between the pollution removed and the crown cover (r = 0.408, $p < 0.001$), indicating that a larger canopy could remove more air pollution when compared to a species with a narrow canopy structure. In this study, *Pterocarpus indicus* Willd. (15.55 kg/tree$^{-1}$/year$^{-1}$) removed the highest amount of air pollution, followed by *M. leucantha* (2.77 kg/tree$^{-1}$/year$^{-1}$), *A. saman* (2.63 kg/tree$^{-1}$/year$^{-1}$), *G. sepium* (2.56 kg/tree$^{-1}$/year$^{-1}$), and *S. roxburghii* (2.27 kg/tree$^{-1}$/year$^{-1}$).

The provisioning service provided (as indicated by timber value) was estimated around a total of USD 84,532, with an average of USD 89.33/tree$^{-1}$/year$^{-1}$. The five species with the highest timber value were *A. indica* (USD 487.73), *S. roxburghii* (USD 376.60), *M. leucantha* (USD 369.48), *H. odorata* (USD 230.26), and *D. cochinchinensis* (USD 229.06) (Table 2). The timber value contribution to the total monetary value was 75%, as most of the tree species selected for planting in the park were of high economic value, which resulted in a high estimate of timber values.

## 4. Discussion

The monetary values of ecosystem services provided by an urban green space converted from a junk yard and shop houses to a park designed to mitigate the ill effects of flooding in a crowded residential area in Bangkok, Thailand were investigated. The monetary estimation of regulating and provisioning services provided by this park was much lower than its cost of construction (approximate USD 0.1 vs. USD 45 million). However, other types of regulating and provisioning services were not included in this study, nor were culturing and supporting services [56], which could increase the monetary value offered by the park. As most ecosystem services are intangible assets, which are difficult to evaluate in terms of monetary value, the method of such monetary estimations are highly dependent on the people and their willingness to pay to maintain the ecosystem. It is to be noted that once an urban green ecosystem is established, their services can last for a long time, as long as no extreme disturbances degrade or destroy the structure and function of the ecosystem. Therefore, the services provided by the ecosystem can also accumulate and last for a long time.

The primary goal of the CU 100 Park—to prevent floods in residential areas around the park—was met, given that the estimated ecosystem services in avoided runoff contributed to over 60% of the regulating services (Figure 3). In addition, after the mega-floods of 2011 in Bangkok and the subsequent establishment of the park in 2016, the surrounding areas have never experienced any flooding, despite the rather flat terrain of Bangkok, which is between 0.5 and 1.5 m above the mean sea level [57]. The impervious surfaces in urban areas can generate more than five times the runoff in a forested area of the same size [58]. In this study, trees helped to reduce the surface runoff in the park by 3%. This estimate seems to be low when compared to the total runoff, but this service should be seen in conjunction with a slope of 3 degrees, or approximately 6% of the area (Figure 1). In the classic runoff experiment conducted by Duley and Hays [59], it was indicated that the avoided runoff would increase by up to 75% when land slope was around 6%. In summary, the avoided runoff services from trees and slopes of the area, with a retention pond to collect the excess water, resulted in effective flood prevention in the CU 100 Park and its surrounding areas.

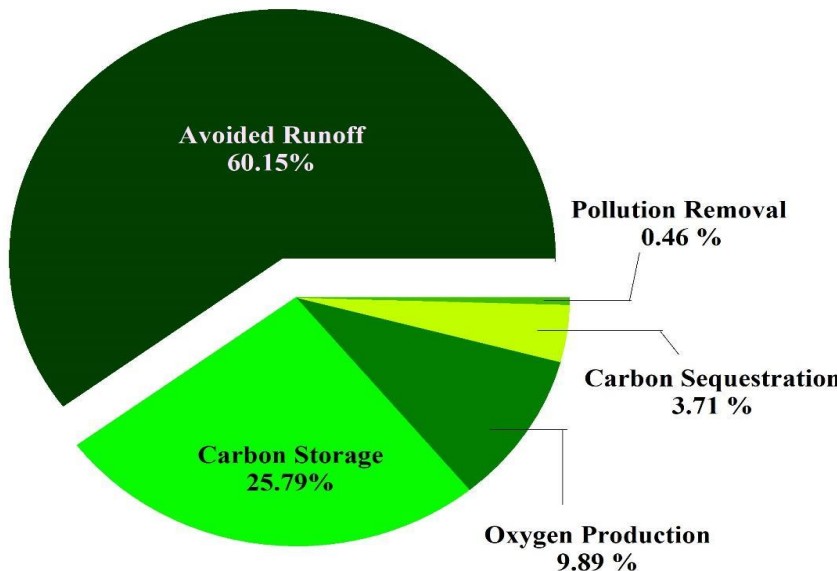

**Figure 3.** Regulating services contributing at least 25% to the total benefits from the park, including avoided runoff, carbon storage, carbon sequestration, oxygen production, and pollution removal.

The main tree traits contributing to the avoided stormwater runoff were the size of the tree and the crown size, which were strongly correlated with the avoided runoff (r = 0.851, $p < 0.001$). The species *M. leucantha* (0.37 m$^3$/tree$^{-1}$/year$^{-1}$) had the highest runoff avoidance due to its large and healthy canopy structure with smooth and widely

spread branches. A complex broad pinnate leaf structure, good growth in full sunlight conditions, and high water demand resulted in the high tolerance of this species to the urban island heat effect. Apart from the size of the stem and crown, which helps to slow down the speed of water from the rain through stem flow, other characteristics that could play a crucial role in contributing to the amount of avoided runoff included the shape of the crown (rounded or umbrella), drought and flood tolerance, high light demand, and whether the species is fast-growing. Urban trees are essential for reducing runoff volumes because they capture precipitation in their canopies, while their root systems can filter and store water in the soil [60].

*4.1. Ecosystem Benefit Value Comparison*

Of all the regulating services that were evaluated, avoided runoff was of the most immediate concern for this park, given that it was specifically designed to mitigate floods with a sloping landscape (Figure 1). We estimated the monetary value of runoff avoidance at $0.177 \text{ m}^3/\text{year}^{-1}$, which is considered relatively low when compared to a similar estimation for the street trees in the Mueng district, Nakhon Si Thammarat Province, Thailand ($37.31 \text{ m}^3/\text{year}^{-1}$) [25]. The lower value estimated in the present study might be due to a smaller tree size range (4.6–30.9 cm) than the trees in Nakhon Si Thammarat Province, which have a range of 15.2–61.0 cm. As tree size can influence the leaf area index, which is strongly related to the avoided runoff, trees with a larger and denser crown cover would provide higher avoided runoff services.

Only one previous study, by Intasen et al. [61], used the i-Tree Eco model to estimate the monetary value of ecosystem services for street trees in 184 stratified random sample plots in Bangkok. Overall, our estimates of the monetary value of the regulating services were relatively higher compared to the study by Intasen et al. [61]. Even the size, height, and crown diameter of trees reported by Intasen et al. [61] were larger than those in this study, with the estimated regulating services, including carbon storage, carbon sequestration, and pollution removal being 43%, 16%, and 6.2% lower, respectively, than those in our study. This difference could be due to the tree density and spacing of planting in a park compared to the street trees. In the present study, tree canopy cover was relatively denser than that reported by Intasen et al. [61], as indicated by 28% (155 trees/ha$^{-1}$) vs. 8.6% (27 trees/ha$^{-1}$), leading to a higher estimate of the total services provided in the present study.

The air pollution removal by trees in the CU 100 Park was slightly lower than that in Fushun, China (619 vs. 740 kg) [62] due to a narrow canopy width, but the value was much higher than that reported by Tan et al. for street trees in Japan [32], as the trees in the CU 100 Park are denser than the street trees in the study in Japan. Apart from plant traits, local weather conditions and the location of the station measuring pollution are also equally important while calculating the model estimates. Szkop [63] indicated that the data from different weather and pollution stations could cause differences of up to 50% in the monetary value estimates. In this regard, although the tree data in our and Intasen et al.'s study [61] was collected from the respective study sites in Bangkok, the weather and pollution data in both the studies came from different locations. This may have resulted in the model underestimating the monetary value in Intasen et al. [61], resulting from different weather and pollution data locations.

According to the Department of National Park, Wildlife and Plant Conservation (DNP)of Thailand [64], the above-ground carbon stocks in evergreen forests is 335,040 kg/ha$^{-1}$/year$^{-1}$, while those in the deciduous forests is 103,680 kg/ha$^{-1}$/year$^{-1}$. In the present study, carbon stored by urban trees was around 7839.29 kg/ha$^{-1}$/year$^{-1}$ (based on 61% evergreen and 39% deciduous species). These values are much less than the values reported by the DNP [64], because the tree size distribution between the urban and natural forests is different, with a majority of large old trees being found in natural forests while relatively younger and smaller trees are more frequently sampled in urban environments [65]. Trees in the CU 100 Park were balled with a diameter between 4.6–30.9 cm and had relatively fast growth rates, given their access to a more open urban forest structure [66]. Moreover,

the species diversity in a natural forest setting is much higher than in a park, leading to different growth rates and biomass accumulations. The density in a natural forest is much higher than that in the park, due to natural regeneration rather than spaced planting in an urban park. Therefore, the estimated biomass and, in turn, carbon storage can be much lower in an urban park compared to in a natural forest. Carbon storage by the trees in the CU 100 Park was massively lower (around seven times lower) than a park in Beijing's built-up area [67], as well as a medium-size planned city in Henan, China [30], due to the smaller-sized and less dense trees in the CU 100 Park, compared to the areas reported in China.

Timber value contributed to 75% of the total monetary value because most of the tree species selected for planting in the park were of high economic value, which resulted in a higher timber value. For example, *D. cochinchinensis*, *S. roxburghii,* and *M. leucantha* are hardwood species of high economic value to the Thai community. Moreover, some of the highest estimated timber values were for the native hardwood species. Stems with high densities and hardwood generally have higher survival rates, due to reasons such as biomechanical and hydraulic safety, pathogen resistance, and physical damage. Moreover, combined with a large size and a high wood density, they play an important role in global carbon storage [43,51].

*4.2. Uncertainty of the Model in Estimating Ecosystem Services*

The estimates of the monetary benefits, in terms of the regulating and provisioning services by the CU 100 Park, include some uncertainties in the assumptions, input data, model parameters, model structures, and the staff who collected the data [68]. If the allometric equations used to estimate the tree growth and biomass in the model [24] were not species-specific, it could introduce an error in the final estimates. The current method used in the i-Tree Eco model was based on a crown-based allometric equation developed using the data collected in Chicago, U.S.A. [69]. In the absence of no species-specific allometric equations, the next level of a taxonomic approach would have to be used, based on the genus or family, to calculate the biomass [69], leading to a higher uncertainty in the estimated biomass. Fortunately, species-specific equations for each of the urban tree species are provided by the i-Tree international software used in the present study, which can be used to reduce this uncertainty [27].

Recently, it has been reported that leaf area can strongly influence the uncertainty [26] because it is mainly used to estimate the amount of air pollution removal and avoided runoff. Therefore, a miscalculation of the leaf area would lead to an uncertainty. Apart from the allometric equations, the locations of weather and pollution stations can also affect the model estimates. Szkop [63] found that nitrogen dioxide ($NO_2$) was the most-affected variable when estimating the ecosystem services in Poland. When applying the different air quality monitoring stations between "urban background" and "traffic background", it was reported that $NO_2$ pollution was 2–3 times higher in traffic background, with the monetary value of air pollution removal being twice the value when the urban background pollution data were substituted by the traffic background data. In Germany, the specific time of the year during a heatwave affected the thermal energy calculation [70]. Nevertheless, despite this uncertainty, data collection and input data can also be errors. Researchers need to carefully plan the data collection and quality control of the input data so that the model can estimate the services with the least possible errors in order to reduce any resulting uncertainties [63].

## 5. Conclusions

The main design of the CU 100 Park was to prevent floods, and 60% of the total regulating services contributed to the "avoided runoff", with the conclusion being that its design goal was achieved. With both evergreen (61%) and deciduous (39%) species growing in the park, regulating and provisioning services were estimated at USD 101,400.60. *A.indica*, *S. roxburghii*, and *M. leucantha* provided the greatest annual benefits, as indicated

by their high value of timber, carbon sequestration, and carbon storage. However, other types of regulating services, such as pollination, pest control, seed dispersal, disease regulation, and erosion control were not measured in this study. In future studies, all such services can be included to estimate the total regulating services to add up to the sum of monetary values. The results reported in this study can be used for policy decisions in the future construction of green spaces in urban areas with unique layouts to maximize the ecosystem services obtained from the park. Planting diverse species with high ecological services as well would enhance human well-being in such urban areas.

**Supplementary Materials:** The following are available online at https://www.mdpi.com/article/10.3390/su132413624/s1, Table S1: Combined monetary benefits of the regulating and provisioning services in the Chulalongkorn Centenary Park (CU 100 Park).

**Author Contributions:** Conceptualization, N.L. and P.T.-N.; methodology, N.L.; validation, P.T.-N.; formal analysis, A.Y. and N.L.; investigation, S.T. and S.H.; data curation, A.Y.; writing—original draft preparation, A.Y. and N.L.; writing—review and editing, A.Y., N.L. and P.T.-N.; visualization, A.Y. and N.L.; supervision, N.L., P.T.-N. and A.P.; funding acquisition, A.Y. All authors have read and agreed to the published version of the manuscript.

**Funding:** This research received no external funding.

**Institutional Review Board Statement:** Not applicable.

**Informed Consent Statement:** Not applicable.

**Data Availability Statement:** Not applicable.

**Acknowledgments:** The authors would like to appreciate the KU forest ecology team for their assistance with field data collection, and the team at the Davey Institute/USDA Forest Service, especially Alexis Ellis and David Nowak, who helped immensely with the usage of the i-Tree Eco International (V6) software.

**Conflicts of Interest:** The authors declare no conflict of interest.

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
