# Peer review of "Evaluation of Regulating and Provisioning Services Provided by a Park Designed to Be Resilient to Climate Change in Bangkok, Thailand"

_sustainability, doi:10.3390/su132413624_

Round 1
Reviewer 1 Report
GENERAL COMMENTS
The paper reports an application of the i-Tree Eco model to study a park in the city of Bangkok. The text is readable, the conclusions are coherent with the obtained results and material and method are well reported. The subject of the study is suitable for the publication on Sustainability.
I suggest the Authors to explicitly explain why they choose the model i-Tree Eco for this study and if there are other suitable models that they are decided not to use.
Author Response
Point 1: I suggest the Authors to explicitly explain why they choose the model i-Tree Eco for this study and if there are other suitable models that they are decided not to use.
Response 1: We have added the explanation in the introduction section according to the suggestion as “Recent development of models has helped with the calculation and estimation of ES. A few examples are ARIES, Co$ting Nature, LUCI, InVEST, Water Word, and i-Tree (Ding and Bullock, 2006; Hamel et al., 2021). However, some models have focus on mapping and land use outside urban areas such ARIES, Co$ting Nature, and LUCI. While some models are available online and are free, such as InVEST and i-Tree Eco but the level of complexity varies in each model. For example, InVEST provides a suite of 18 models as tools to estimate the value of different services (Hamel et al., 2021)”. Currently in line 71-77 (highlight in yellow)

Reviewer 2 Report
Analysis and evaluations of ecosystems services is in recent time very demanded from whole society. Proposed topics I consider as very eligible. Authors presented a Case study from some hectares large Park, situated in urbanized area in centre of Bangkok, Thailand. Complete trees inventory in 2019 were combined with available the i-Tree Eco model (USDA, USA). Authors presents very simply results of Carbon storage, Carbon Sequestration, Avoid Runoff, Pollution Removal and Oxygen production, obtained from this model, including monetary benefits in US dollars.
I ´m quite satisfied with presented study. By my opinion is Abstract succinct, Introduction well-combined, chosen design described in Material and Methods appropriated. Results and briefly Discussion are transparent and suitable. Authors by my opinion have used relevant literature and appropriate citations.
I have only few comments or questions to authors to improving final quality of this Article:
Row 60 contain reference Wolf 2004 – modify text to number according MDPI requirements
I recommend improve Quality of Figure 1, right part is little-bit hardly readable, please append in Title comments closer explained (a) and (b) part.
In chapter Methods authors presents, that all trees were measured one-time, during September and October 2019 (row 141). In next text (row 174-175) presents, that the annual carbon sequestration was calculated as the difference between the carbon storage during the current and the next year. How was annual carbon sequestration really derived? Please, explain concrete practice.
Repair page numbering from page 6 (currently is from page 6, bad numbering o the end) .
Enlarge text size in table 1, current size is very hard readable.
Too text size in Figure 2 is very small and unreadable.
Reference 40 and 41 is same, please renumber next references and repair numbering in text - I think that reference 42 is bad cited, but from number 44 in text I expect next references as correct. Please, check and correct whole text.
After thereinbefore modifications I recommend publish this paper after little modifications as Article in Sustainability Journal.
Author Response
Point 1: Row 60 contain reference Wolf 2004 – modify text to number according MDPI requirements
Response 1: All the references have been carefully checked throughout the manuscript.
Point 2: I recommend improve Quality of Figure 1, right part is little-bit hardly readable, please append in Title comments closer explained (a) and (b) part.
Response 2: Figure 1 has been updated with the higher resolution and clearer explanation.
Point 3: In chapter Methods authors presents, that all trees were measured one-time, during September and October 2019 (row 141). In next text (row 174-175) presents, that the annual carbon sequestration was calculated as the difference between the carbon storage during the current and the next year. How was annual carbon sequestration really derived? Please, explain concrete practice.
Response 3: Carbon sequestration is calculated from the model based on the study of Nowak and Crane (2002) who estimated the difference between carbon storage in the current year (X) and the next year (X+1). In general, carbon storage is estimated using the allometric equation between tree size (measured as diameter at breast height: DBH) and dry biomass. Therefore, by measuring only DBH at the point of trees sampling, as explained in the materials and method section, we would be able to estimate the carbon storage during the current year. Then, the carbon storage of the next year (X+1) could be estimated using the growth rate of a species. This was based on the growth estimation by Nowak (1994), who indicated that the average DBH growth of trees in park-like structures (e.g. parks, cemeteries, golf courses) was about 0.38 cm/year and was 0.61 cm/year for more open-grown trees. The carbon storage for the next year (X+1) is calculated using the estimated DBH growth rate along with tree conditions (such as tree health) that were measured during this year. Using these estimates, the difference in the carbon storage of the current year and the next year was calculated. In addition, the rate of mortality and decay was marked as a tree condition to estimate the carbon returned to the air. In summary, the net carbon sequestration was calculated as the difference between the carbon sequestered by the trees and the carbon emitted by the dead parts.
This is the reason why we measured only the DBH only one time and could calculate the carbon sequestration using he equations and estimations mentioned above.
Point 4: Repair page numbering from page 6 (currently is from page 6, bad numbering o the end).
Response 4: We have difficulty problem to find the numbering pages, please kindly assist.
Point 5: Enlarge text size in table 1, current size is very hard readable.
Response 5: Text in Table 1 has been enlarged according to the suggestion.
Point 6: Too text size in Figure 2 is very small and unreadable.
Response 6: Text size in Figure 2 has been enlarged increase the readability according to the suggestion.
Point 7: Reference 40 and 41 is same, please renumber next references and repair numbering in text - I think that reference 42 is bad cited, but from number 44 in text I expect next references as correct. Please, check and correct whole text.
Response 7: All references have been carefully checked throughout the manuscript.

Reviewer 3 Report
Introduction: please explain in a clear way the innovative aspects of your study
Figure 1: add scale bar and improve sharpness (this last comment for all figures
Abstract: please highlight some numerical results also in the abstract Introduction: please provide also in a clear way the research hypothesis at the base of your study All the text: please add the name of the discover to the latin name of the various speciesAuthor Response
Point 1: Introduction: please explain in a clear way the innovative aspects of your study
Response 1: We have added the explanation in the introduction section according to the suggestion as
“Some studies in Asia have reported the use of the i-Tree International (Na et al. 2014; Gonzales and Magnaye, 2017; Song et al. 2020; Cristiano et al. 2020; Tan et al. 2021). Our investigation is the first of its kind undertaken in Thailand in quantifying the monetary value of regulating services using the i-Tree Eco International model of each tree species. The accuracy of estimation would increase due to the use of specific allometric equation and weather data obtained from local stations.” Currently in line 95-99 (highlight in yellow)
Point 2: Figure 1: add scale bar and improve sharpness (this last comment for all figures
Response 2: Figure 1 has been improved according to the suggestion.
Point 3: Abstract: please highlight some numerical results also in the abstract
Response 3: Monetary estimates have been added according to the suggestion.
" Azadirachta indica A.Juss (US$518.75 tree-1 year-1), Shorea roxburghii G.Don (US$417.17 tree-1 year-1) and Millettia leucantha Kurz (US$414.87 tree-1 year-1)" Currently in line 31-34 (highlight in yellow)
Point 4: Introduction: please provide also in a clear way the research hypothesis at the base of your study
Response 4: We have added the explanation in the introduction section according to the suggestion as
“We hypothesized that the regulating services might provide more monetary value relative to the provisioning services due to the complexity of services provided.” Currently in line 120-122 (highlight in yellow)
Point 5: All the text: please add the name of the discover to the latin name of the various species
Response 5: All tree species have been carefully checked and author names have been added throughout the manuscript.